# Descriptive Observational Study of Tdap Vaccination Adhesion in Pregnant Women in the Florentine Area (Tuscany, Italy) in 2019 and 2020

**DOI:** 10.3390/vaccines9090955

**Published:** 2021-08-26

**Authors:** Benedetta Bonito, Daniela Balzi, Sara Boccalini, Paolo Bonanni, Giovanna Mereu, Maria Grazia Santini, Angela Bechini

**Affiliations:** 1Department of Health Sciences, University of Florence, 50134 Florence, Italy; benedetta.bonito@unifi.it (B.B.); sara.boccalini@unifi.it (S.B.); paolo.bonanni@unifi.it (P.B.); 2AUSL Toscana Centro, 50122 Florence, Italy; daniela.balzi@uslcentro.toscana.it (D.B.); giovanna.mereu@uslcentro.toscana.it (G.M.); mariagrazia.santini@uslcentro.toscana.it (M.G.S.)

**Keywords:** COVID-19, pandemic, pregnancy, newborn, pertussis, vaccination, vaccination coverage, Florence, Italy

## Abstract

*Background*: Tdap (Tetanus-Diphtheria-acellular Pertussis) vaccination is nowadays a worldwide-recommended practice to immunize pregnant women. The vaccine administration at the third trimester of pregnancy (as recommended by the WHO) would ensure antibody protection to both the mother and the newborn and has contributed to the significant drop of pertussis cases in infants. The aim of this observational study was to describe for the first time the socio-demographic characteristics and determinants of Tdap vaccination adhesion of pregnant women in the Florentine area. *Methods*: Information about parents’ vaccination status, their citizenship, employment type and mothers’ previous pregnancies and/or abortions were collected at the time of birth through the assistance birth certificates (CedAP) both for the years 2019 and 2020. This archive and the regional SISPC (Collective Prevention Healthcare Information System) linked using an anonymous unique personal identifier to retrieve the mother’s vaccination status. *Results*: We found an overall Tdap vaccination adhesion of 43% in 2019 and 47.3% in 2020. Several socio-demographic parameters would determine an increased vaccination adhesion, including parents’ geographical origin, mothers’ age and educational background, as well as the number of previous deliveries, abortions or voluntary termination of pregnancy. *Conclusions*: Since not much data are available on this topic in Italy, this study may constitute the baseline information for Tdap vaccination adhesion in pregnant women in the Florentine area (Italy). Thus, future successful vaccination strategies may be designed accordingly.

## 1. Introduction

Pregnancy represents one of the most important, crucial hallmarks in the life of a woman, and it may represent a time during which concerns and fears about the safety of vaccines may arise. As a matter of fact, immunization of pregnant women is now a worldwide recommended practice with a double positive effect. First, vaccination directly protects women against infectious diseases particularly dangerous in pregnancy (e.g., influenza); secondly, it also protects the newborns against diseases, such as tetanus and pertussis, until they develop a protecting level of antibodies [1,2]. In particular, pertussis (whooping cough) is a respiratory tract infectious disease caused by *Bordetella pertussis*. It is mostly asymptomatic or mildly symptomatic in older previously vaccinated subjects; on the contrary, it can be particularly severe in infants under 6 months of age, who are too young to have received their primary dose of TDaP (Tetanus-Diphtheria-acellular Pertussis) vaccine and have the highest hospitalization and complication rates [3]. The disease incubation period is typically 9–10 days (range 6–20 days), and the classic clinical presentation in children is paroxysms of coughing, ending with the characteristic whoop and emesis (vomiting). The prognosis can be particularly severe in the first and second year of life, when incidence, as well as hospitalizations and case fatality rates (CFRs), are particularly high (CFRs: 0.2% and 4% in developed and developing countries, respectively) [4]. Deaths primarily occur for pulmonary complications.

The practice to vaccinate pregnant women started back between the 17th and 18th centuries in Asia and in Africa by inoculating smallpox to prevent the infection; since then, this practice was extended to Europe and America [5]. In the 1930s and 1940s, clinical trials on the first whole-cell pertussis vaccine in pregnancy demonstrated safety for the newborn and the potential to transfer protective levels of antibodies to the neonate, reporting no reactogenicity [6]. Pertussis is a worldwide endemic/epidemic infectious disease, reporting outbreaks usually every 2–5 years with a spring-summer seasonality [4]. In 2018, infants <1-year-old were the most affected age group by pertussis in Europe (except for Estonia and Norway), and three deaths were reported. The highest rates of pertussis in infants were reported in Austria (180.4 cases/100,000), followed by Iceland (146.6), Slovenia (123.6) and Denmark (121.5). Infants represented 6% of all cases reported; among those in which the month of age was known (87%), 65% were under six months old, and 45% were under three months old [3]. Moreover, Germany, the Netherlands, Norway, Spain and the United Kingdom altogether accounted for 72% of all notified cases in all age groups. In Italy, it has been reported a rising trend of pertussis hospitalizations in the period 2002–2016, and infants <1-year-old showed the highest percentage (63.4%) and average hospitalization rates (74.6/100,000) [7]. During the period 2010–2019, pertussis vaccination coverage (VC) rates in the Tuscany region were reported as >95% (except for the year 2016); nevertheless, 38 cases of pertussis were notified in Tuscany in 2019, and 11 of these were hospitalized [8]. Importantly, it is known that mothers are the main source of infection for their babies within the households (39%), then fathers (16%) and grandparents (5%) [9]. It is, therefore, extremely important to immunize family members and careers too (cocoon strategy) in order to protect the unvaccinated newborns. According to this evidence, it is primarily critical to vaccinate pregnant women against diphtheria-tetanus-pertussis (Tdap), and hence, the Italian National Immunization Plan 2017–2019 (NIP) recommends giving the Tdap vaccination at the 28th week of pregnancy at each pregnancy, regardless of previous Tdap doses received [10]. Moreover, the Italian Ministry of Health endorsed two documents (in 2018 and in 2019), including recommendations on vaccinations for pregnant women [11,12]. Thus, based on international [13], national [11,12] and regional guidelines [14], the Local Health Unit Toscana Centro (LHU-TC) actively promoted Tdap vaccination in pregnant women, starting from the training of all the health professionals involved.

The aim of this observational study was to describe the socio-demographic characteristics and determinants of Tdap vaccination adhesion of pregnant women in the LHU-TC during the year 2019 and compare the vaccination adhesion to the year 2020.

## 2. Materials and Methods

The Tdap promotion vaccination guidelines in the LHU-TC were directed to: gynecologists and pediatricians (hospital and private practice), GPs, hygienists, public health professionals, healthcare assistants and medical doctors working in the vaccination centers. Particular attention was given to the midwives’ category, which was trained by the Public Health Unit professionals. An informative vaccination booklet to be given to the women was also distributed to all the specialists working in the Healthcare Centers of the LHU-TC and to the GPs. In order to consistently promote vaccinations to pregnant women, all the professionals mentioned followed a common procedure and provided all the information at the delivery of the pregnancy booklet (which includes all the recommendations about Tdap vaccination) [15], at the pre-natal screening visits and during the routine mothers’ visits (including the post-partum period). Pregnant women could receive the vaccination at the vaccination centers (even with no booked appointment), at their GP’s, at the time of birth and even at the post-partum check, although at this time, vaccine efficacy may be reduced. For the purpose of this study, information about the parents’ vaccination status, mother’s age and geographical area of residence in the LHU-TC, their citizenship, employment type and the mothers’ previous pregnancies and/or abortions were collected at birth time through the assistance birth certificates (CeDAP). This archive and the regional SISPC (Collective Prevention Healthcare Information System) (Consortium Metis, Tuscany, Italy) were linked using an anonymous unique personal identifier to retrieve mother’s vaccination status.

Data were collected during the years 2019 and 2020, before and during the COVID-19 pandemic emergency. The description of the socio-demographic characteristics of the pregnant women resident in the LHU-TC was referred to the year 2019 database. The geographical area of origin was reported divided by the Ex LHU of residence: Pistoia, Prato, Florence, Empoli. The population enrolled in the year 2020 in the same geographical area accounted for 9652 subjects and presented comparable socio-demographic characteristics to those enrolled the year before. Tdap vaccination adhesion in pregnant women was retrieved for both 2019 and 2020 years.

Statistical significance was assessed with the chi-squared (χ^2^) test, which was used to compare relative frequencies (assuming a *p*-value < 0.05 as statistically significant).

## 3. Results

Our sample population for the year 2019 included 10,063 women (*N*) resident in the LHU-TC. The social and demographic characteristics of the participants are shown in Table 1.

### 3.1. Tdap Vaccination Adhesion by Geographical Area of Residence, Age Group and Parents’ Citizenship

Overall, the percentage of Tdap vaccinated women in our sample population in 2019 was 43.0% (4328/10063). Women resident in Prato showed the highest vaccination percentage (50.8%), followed by those resident in Pistoia (44.5%), Florence (42.9%) and Empoli (33.4%) (*p* < 0.05). Figure 1 shows the distribution of Tdap vaccinated women according to their age group. The most vaccinated were women belonging to the 35–39 age group (48%, 1380/2874), whereas the least were in the 18–24 age group (22.7%, 173/762) and in the 50–54 (30.8%, 4/13) (Figure 1). The chi-squared test reported to be statistically significant (*p* < 0.05).

Table 2 summarizes the Tdap vaccination status of the pregnant women by mother’s and father’s citizenship. Italian pregnant women (who are also the most represented in our sample) showed the highest vaccination rates: 51.1% among Italian mothers (3592/7034) and 49.4% among those having Italian partners (2611/5288). Moroccan, Senegalese and Nigerian mothers were those with the lowest Tdap vaccination adhesion: 10.4% for Morocco and 14% for pregnant women having Moroccan partners; 13.5% for mothers of Senegalese citizenship and 11.6% for pregnant women having Senegalese partners and 15.3% for Nigerian women and 14.5% for those having Nigerian partners. For the 45.5% of the vaccinated women (1233/2712), it was not possible to retrieve information about partners’ citizenship (*p* < 0.05).

### 3.2. Tdap Vaccination Adhesion Analysis by Parents’ Educational and Occupational Status

We then retrieved information about mothers’ marital status and both parents’ educational profiles. Not married women had the highest Tdap vaccination rate (46.8%), followed by those married (40.2%); the separated and the divorced vaccinated women were 27.7% and 37.7%, respectively; for almost 42% of them, this information was not detected. Moreover, none of the widowed women were vaccinated (0/4). The chi-squared test applied to mothers’ marital status was statistically significant (*p* < 0.05). Looking at the educational background, it appears that the vaccination adhesion trend would go hand in hand with mothers’ academic qualifications: 55.0% of women who held a first-level degree (bachelor’s degree) were vaccinated, 50.8% of those holding a second-level degree (master’s degree), 42.1% of women holding a high school certificate, and finally, 33.1% of vaccinated women had a middle school qualification; for 38% of the vaccinated women, this information was not detected. The lowest vaccination rate was represented by women who did not hold any scholastic qualification (14.1%) (*p* < 0.05). Tdap vaccination adhesion rates of pregnant women are comparable when considering the fathers’ academic qualification: the majority of them (51.6%) had a partner with a master’s degree, 44.8% had a partner with a high school certificate, 44.4% had a partner with a bachelor’s degree, 36.3% had a partner with a middle school certificate, and lastly, 13.4% of vaccinated women had a partner with no qualifications. For almost half of the vaccinated pregnant women (44.8%), information about their partners’ qualifications was not available (*p* < 0.05). Furthermore, employed mothers reflected a high rate of Tdap vaccinated women (49.6%) compared to those unemployed (35.5%). The chi-squared analysis gave a statistically significant outcome (*p* < 0.05).

### 3.3. Analysis of Previous Pregnant Status, Deliveries and Voluntary Termination of Pregnancy (VTP)

Only a small percentage of women who had previously conceived had received Tdap vaccination (almost 37%) in 2019, while 52.3% among those expecting their first child were vaccinated. More than half of the women who were delivering for the first time received a Tdap dose (51.9%). The percentage of vaccinated women decreased by the number of previous deliveries: 39.5% (one delivery), 31.9% (two deliveries) and 26.4% (three or more deliveries) (Figure 2). Tdap vaccination adhesion in those who had previously aborted spontaneously is 43% (the lowest rate at 26.3% for ≥4 abortions); on the other hand, the Tdap vaccination trend in those who aborted with a voluntary termination of pregnancy (VTP) decreases overall by the number of VTPs. The Chi-squared test was highly statistically significant for VTPs and the number of previous deliveries (*p* < 0.01 and *p* < 0.001, respectively). However, it was not significant for the number of previous abortions.

Only the 5% of women in our sample (506/10063) were medically assisted to begin the pregnancy, and 56.5% of them got a Tdap vaccine dose, whereas 43.5% of them were unvaccinated. We also considered the number of medical visits during pregnancy and, those women who had up to four visits showed an average vaccination rate of 24.9% (compared to the overall 43.0%); whereas women who had a number of visits between 5 and ≥13 during their pregnancy showed a vaccination rate of 43.9% (*p* < 0.05).

### 3.4. Comparison of Tdap Vaccination Adhesion in 2019 and 2020

We also retrieved Tdap vaccination adhesion for the year 2020. We found an overall slightly increased trend rising from 43% in 2019 to 47.3% in 2020. Florence and Empoli were the geographical areas that registered the greatest Tdap vaccination increase in these two years: from 42.9% to 47.5% and 33.4% to 45.7%, respectively. The area of Prato was confirmed to be the top one, with 51.4% of vaccinated women (50.8% in 2019). Overall, women appeared to be more vaccinated in the year 2020 compared to 2019 for the following socio-demographic characteristics, with a comparable pattern for each analyzed parameter: parents’ citizenship (*p* < 0.0001), mothers’ marital status (*p* < 0.0001), fathers’ educational level (*p* < 0.0001), mothers’ professional occupation (*p* < 0.05), medically assisted procreation (*p* < 0.05) and the number of visits during pregnancy (*p* < 0.0001).

## 4. Discussion

Despite the excellent pertussis VC at 24 months (>95% both in Italy and Tuscany region), a few cases of whooping cough in infants still occur [8,16]. The most effective way to keep protecting the newborns from pertussis is to vaccinate pregnant women with a dose of Tdap in their third trimester of pregnancy to ensure antibody protection to the baby. The aim of this study was to describe the socio-demographic characteristics and determinants of Tdap vaccination of pregnant women in the LHU-TC during the year 2019 and compare the vaccination adhesion to the year 2020. Our sample population enrolled 10,063 pregnant women in 2019, and this well represented the number of newborns in the area of Pistoia (1831), Prato (1752) and the area of Florence (5136) and Empoli (1545), giving a total number of 10,264 babies [17]. Women between 35–39 years of age showed a higher vaccination percentage (48% in 2019 and 52.4% in 2020) than those younger (i.e., 22.7% 18–24 years old in 2019 and 26.1% in 2020). According to the NIP 2017–2019, the best time for Tdap vaccination during pregnancy is from the 27th to the 36th week of gestation [10]. Despite these recommendations, however, the Tdap VC among pregnant women remains very low not only in Italy but also worldwide [18,19]. In Southern Italy, the Tdap vaccine adhesion in pregnant women is about 1.7% [20], in the Abruzzo region between 1.2–15% (for both influenza and Tdap or only one of them) [21], whereas in countries, such as Belgium, it reached the 64% [22,23]. The main reason for women’s refusal appears to be the lack of appropriate information provided by the GP and obstetricians and/or gynecologists [24]. Unaware or poorly aware pregnant women were more likely to feel that the recommended vaccines administered during pregnancy (Tdap and influenza) were very dangerous for them and for their unborn child [25,26,27]. The cultural and demographic background also represents a great variable in the access to vaccination. Indeed, Moroccan, Senegalese and Nigerian mothers showed the lowest level of Tdap vaccination amongst those enrolled in our study, and the same results were observed when considering the father’s citizenship. Since the 1980s, only a few pertussis cases have been reported in Morocco [28] and Senegal [29]; however, Nigeria seemed to be having a great number of cases in the last 40 years, peaking at 92.266 pertussis cases in the year 1985 [30]. Mothers’ academic level, as well as employment type, increased the vaccination awareness and adhesion greatly, as is also widely reported in the literature [31,32]. Having a high educational background might facilitate a pregnant woman’s communication with physicians and greater accessibility to different sources and ability to understand and interpret the received information. The year of the SARS-CoV-2 outbreak registered an increase of Tdap vaccination adhesion in pregnant women, rising from 43.0% in 2019 to 47.3% in 2020. The pandemic has indeed increased women’s perceptions and fears about childbirth [33,34]. Thus, women may have chosen to be immunized against preventable infectious diseases to feel safer for themselves and their child.

## 5. Conclusions

Our study shows for the first time the level of Tdap immunization in pregnant women within the Florence area (Italy). This screenshot highlights how, although the several promotion campaigns adopted in the territory, pregnant women appear not to be fully aware of the vaccination’s beneficial effect both for themselves and for their babies. The lack of information provided by physicians and healthcare staff seems to be the main reason why women expecting a child refuse to get the vaccination, being worried about the safety of vaccines. Thus, since the Tdap vaccination is the only preventive available tool to immunize newborns, it is strongly necessary to improve the training of the healthcare workers (gynecologists, midwives, pediatricians, GPs, healthcare workers, health assistants and medical doctors) to ensure a high and efficient level of information is provided to future mothers. Moreover, pertussis prevention in infanthood is based not only on the mothers’ immunization programs but also on keeping the VC rates high (95%) in all age groups. It is therefore important to administer Tdap booster doses in adults every ten years. The goal is to reduce significantly the circulation of *Bordetella pertussis* within the population. Finally, this study could be used as a baseline information tool to design future successful vaccination strategies targeted to pregnant women.

## Figures and Tables

**Figure 1 vaccines-09-00955-f001:**
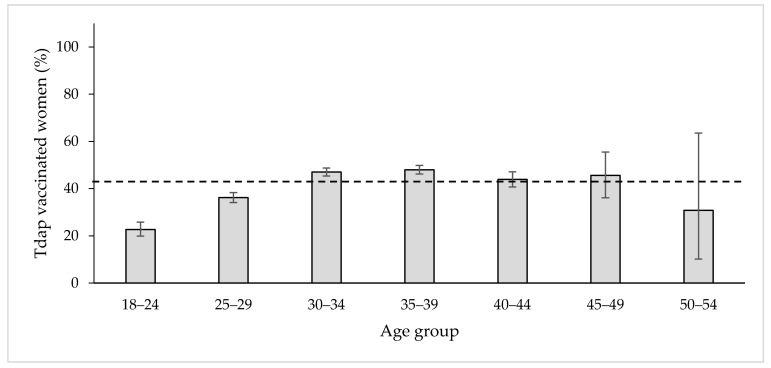
The percentage distribution of Tdap vaccinated women in 2019 according to the age group. The dashed line indicates the average percentage adhesion: 43.0%.

**Figure 2 vaccines-09-00955-f002:**
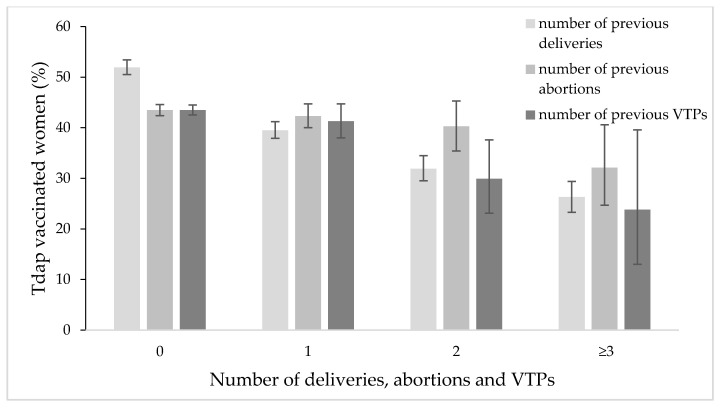
Percentage distribution of Tdap vaccinated women by number of previous deliveries, abortions and VTPs.

**Table 1 vaccines-09-00955-t001:** Socio-demographic characteristics of the sample population enrolled in 2019. Frequency (%) = *n*/*N*.

Total Number (*N*) = 10,063				
Socio-Demographic Characteristics	*n*		%	
**Ex LHU of residence**				
Pistoia	1758		17.5	
Prato	1631		16.2	
Florence	5130		51.0	
Empoli	1544		15.3	
Age				
18–24	762		7.1	
25–29	1969		18.8	
30–34	3407		35.2	
35–39	2874		28.3	
40–44	934		9.4	
45–49	103		1	
50–54	13		0.1	
**Citizenship**	**Mothers**	**Fathers**	**Mothers**	**Fathers**
Italy	7034	5288	69.9	52.5
Albany	619	519	6.2	5.2
Romania	354	194	3.5	1.9
China	559	470	5.6	4.7
Morocco	259	207	2.6	2.1
Peru	141	53	1.4	0.5
Nigeria	85	62	0.8	0.6
Senegal	88	69	0.9	0.7
Other	924	489	9.2	4.9
N.D.	-	2712	-	27.0
**Mothers’ marital status**				
Married	4666		46.4	
Unmarried	4178		41.5	
Separated	65		0.6	
Divorced	69		0.7	
Widowed	4		0.0	
N.D.	1081		10.7	
**Educational level**	**Mothers**	**Fathers**	**Mothers**	**Fathers**
None	128	134	1.3	1.3
Middle school	2180	2535	21.7	25.2
High school	3885	3171	38.6	31.5
Bachelor’s degree	1968	288	19.5	2.9
Master’s degree	1166	1130	11.6	11.2
N.D.	737	2805	7.3	27.9
**Occupational status**				
Employed	6635		65.6	
Unemployed	1201		12.0	
Looking for first job	34		0.3	
Housewife	1513		15.3	
Student	59		0.6	
Other	604		6.0	
N.D.	17		0.2	
**Previous conceptions**				
Yes	6091		60.5	
**No. previous deliveries**				
0	4513		44.9	
1	3424		34.0	
2	1309		13.0	
3	503		5.0	
4	193		1.9	
5	66		0.7	
≥6	55		0.5	
**No. previous abortions**				
0	7860		78.1	
1	1677		16.7	
2	380		3.8	
3	96		1.0	
≥6	38		0.4	
N.D.	12		0.1	
**No. previous VTP (voluntary termination of pregnancy)**				
0	9037		89.8	
1	816		8.1	
2	154		1.5	
≥3	42		0.4	
N.D.	14		0.1	
**Medically assisted**				
**Procreation**				
Yes	506		5.0	
**No. visits during pregnancy**				
0	6		0.1	
1	134		1.4	
2	120		1.2	
3	325		3.2	
4	616		6.1	
5	1609		16.0	
6	1788		17.8	
7	1556		15.5	
8	1811		18.0	
9	693		6.9	
10	700		7.0	
11	189		1.9	
12	199		2.0	
≥13	285		2.8	
N.D.	32		0.3	

**Table 2 vaccines-09-00955-t002:** Tdap vaccination adhesion in pregnant women by mothers’ and fathers’ citizenship.

Vaccination Adhesion in Pregnant Women		
Citizenship	Mothers% (*n*/*N*)	Fathers% (*n*/*N*)
Italy	51.1 (3592/7034)	49.4 (2611/5288)
Albany	26.5 (164/619)	26.3 (137/519)
Romania	24.0 (85/354)	21.6 (42/194)
China	36.0 (201/559)	36.2 (170/470)
Morocco	10.4 (27/259)	14.0 (29/207)
Peru	25.4 (36/141)	20.8 (11/53)
Nigeria	15.3 (13/85)	14.5 (9/62)
Senegal	13.5 (11/88)	11.6 (8/69)
N.D.	-	45.5 (1233/2712)
Other	21.5 (199/924)	15.0 (78/489)

## Data Availability

The data presented in this study are available on request from the corresponding author. The data are not publicly available due to privacy reasons of LHU policies.

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
