# Peer review of "Descriptive Observational Study of Tdap Vaccination Adhesion in Pregnant Women in the Florentine Area (Tuscany, Italy) in 2019 and 2020"

_vaccines, 2021, doi:10.3390/vaccines9090955_

Round 1

Reviewer 1 Report

Descriptive Observational Study of Tdap Vaccination 2 Adhesion in Pregnant Women in the Florentine Area 3 (Tuscany, Italy) in 2019 and 2020

This study is interesting. I have detected these small minor changes in the manuscript. There are some statistical aspects, which should be correct it before the final acceptation. Please, remove ages from 17-19 years old in all tables since n=9 is not representative here.

My Decision is Minnor revision

Commenst to the Editor and authors

Minnor

-Why these authors indicate data from The highest rates of pertussis in infants 60 were reported in Austria (180.4 cases/100 000), followed by Iceland (146.6), Slovenia 61 (123.6) and Denmark (121.5). Infants represented 6% of all cases reported; among those in 62 which the month of age was known (87%), 65% were under six months old and 45% were 63 under three months old [3]. Which is the reason of the lack of data from more representative European countries in vaccunation? Are important these data from Slovenia in terms of vaccunation?

-In my opinion, these data from 17-19 years old should be removed from the table given the extremely low size sampe (n=9) and also remove data from N.D in this table (also in table 2: remove it N.D data)

-Please, include S.E.M (error standard media) in graph of percentages and also remove the age group between 17-19 years old.

-Include S.E.M (error standard media in tfigure 2, which is the variance divided by roof of n, being n the size sample and put * in case of significance between groups.

Thanks¡

Author Response

Please find the authors' replies attached. 

Thank you

Reviewer 2 Report

In this manuscript, the authors study the adhesion to Tdap vaccination of pregnant women in Florentine area according several socio-demographic parameters. The work is interesting and shows a clear evolution from previous study in Italy before government recommendations (ref 20).

There are several minor points which can be amended.

  1. There are several missing data, mentioned in lines 143, 150 and 157 though in Table 1 most of the information has been collected with low proportion of N.D. category. What is the origin of the missing data? Is it a problem with consent of personal data collection?
  2. The proportion of vaccinated women with specific information not detected is important. Does is have an influence on the conclusions drawn with vaccinated women with specific information collected?
  3. The authors could discuss the impact of vaccination recommendations. The study cited in ref 20 was done before such recommendations and the vaccination level among pregnant women was lower.
  4. Small typos errors:

- In table 1, there are shifts in lines alignment for No of previous VTP and No. of visits during pregnancy.

- Line 112. “While Tdap vaccination…”. “While” seems dispensable.

- Line 201 “CV” instead of “VC”

- line 239, a word is missing before “the Florence area”.

Author Response

(The authors gave the same response as above.)
